# Dopamine and Norepinephrine Tissue Levels in the Developing Limbic Brain Are Impacted by the Human *CHRNA*6 3′-UTR Single-Nucleotide Polymorphism (rs2304297) in Rats [note 1]

**DOI:** 10.3390/ijms25073676

**Published:** 2024-03-26

**Authors:** Diana Carreño, Antonella Facundo, My Trang Thi Nguyen, Shahrdad Lotfipour

**Affiliations:** 1Department of Emergency Medicine, University of California, Irvine, CA 92697, USA; 2Department of Pharmaceutical Sciences, University of California, Irvine, CA 92697, USA; 3Department of Pathology and Laboratory Medicine, University of California, Irvine, CA 92617, USA

**Keywords:** nicotine-induced reinstatement, nucleus accumbens, DA turnover, adolescence, addiction

## Abstract

We previously demonstrated that a genetic single-nucleotide polymorphism (SNP, rs2304297) in the 3′ untranslated region (UTR) of the human *CHRNA*6 gene has sex- and genotype-dependent effects on nicotine-induced locomotion, anxiety, and nicotine + cue-induced reinstatement in adolescent rats. This study aims to investigate how the *CHRNA*6 3′-UTR SNP influences dopaminergic and noradrenergic tissue levels in brain reward regions during baseline and after the reinstatement of drug-seeking behavior. Naïve adolescent and adult rats, along with those undergoing nicotine + cue reinstatement and carrying the *CHRNA*6 3′-UTR SNP, were assessed for dopamine (DA), norepinephrine (NE), and metabolites in reward pathway regions. The results reveal age-, sex-, and genotype-dependent baseline DA, NE, and DA turnover levels. Post-reinstatement, male α6^GG^ rats show suppressed DA levels in the Nucleus Accumbens (NAc) Shell compared to the baseline, while nicotine+ cue-induced reinstatement behavior correlates with neurotransmitter levels in specific brain regions. This study emphasizes the role of *CHRNA*6 3′-UTR SNP in the developmental maturation of the dopaminergic and noradrenergic system in the adolescent rat brain, with tissue levels acting as predictors of nicotine + cue-induced reinstatement.

## 1. Introduction

The initiation and establishment of tobacco product use predominantly occurs during adolescence [1]. E-cigarettes have been the most used tobacco product among adolescents since 2014. In 2023, about 4.6% of middle school and 10% of high school students reported using e-cigarettes in the past 30 days [2]. The utilization of tobacco products during adolescence heightens the likelihood of lifelong nicotine addiction and adverse health consequences. There are many factors associated with youth tobacco product use, including social, environmental, cognitive, and genetic influences. Understanding the factors contributing to nicotine use in adolescents is crucial for developing effective prevention and intervention strategies. 

Exposure to nicotine during adolescence disrupts normal neurochemical functions, affecting nicotinic acetylcholine receptor (nAChR) subunits and the neurotransmitter content related to reward, with potential persistence into adulthood [3,4]. Nicotine binds to nAChRs composed of alpha (α)2–10 and beta (β)2–4 subunits in neuronal cells. Nicotine facilitates phasic firing and increased dopamine (DA) and norepinephrine (NE) release. As a result, neurotransmission levels are heightened in limbic brain regions supporting the reinforcing effects of nicotine [5]. Brain regions most impacted by nicotine include the prefrontal cortex (mPFC), dorsal caudate putamen (dCPu), Nucleus accumbens (NAc), basal lateral amygdala (BLA), interpeduncular nucleus (IPN), ventral tegmental area (VTA), and locus coeruleus (LC), as some examples [6]. The research suggests variations of the *CHRNA*6 gene may play a role in modulating DA and, to an extent, NE release. Neuronal nAChRs containing the α6 subunits are expressed on DA-releasing neurons in the brain [7,8]. In addition, nicotine-induced DA release has been characterized using subtype selective ligands, antibodies, and nAChR receptor subunit-null mice [9,10,11,12,13]. Further, the LC neurons are a source of ascending brain noradrenergic innervations involved in the regulation of alertness and emotional arousal control which express various nAChRs’ subunits including α6 [14,15,16,17]. To understand the genetic mechanisms mediating these effects, human association studies have indicated that a single-nucleotide polymorphism (SNP) in the 3′-untranslated region (UTR) of the α6 nAChR subunit gene (*CHRNA*6) is associated with increased cigarette smoking, drug experimentation during adolescence, and nicotine dependence and unsuccessful quit attempts in adulthood [18,19,20,21]. The majority of these studies have identified the GG risk allele as the primary genotype associated with enhanced tobacco/nicotine problems, while the CC, non-risk allele is not [7,8,9,11,12,13,14,15,22,23,24,25,26]. We have established the *CHRNA*6 3′-UTR SNP rat line, generating α6^GG^ and α6^CC^ allele carriers [27]. Prior research showed that the *CHRNA*6 3′-UTR SNP knock-in enhances sub-chronic nicotine-induced locomotion, anxiolytic behavior, and the nicotine plus cue-induced reinstatement of drug-seeking behavior in a sex- and genotype-dependent manner in adolescents [27,28]. Mechanisms mediating nicotine-induced behavioral responses in our *CHRNA*6 3′-UTR SNP during adolescence are unknown. 

Our current study assesses tissue-level DA, NE, DA metabolites (3,4-dihydroxyphenylacetic acid (DOPAC) and homovanillic acid (HVA) and turnover rates in the brain regions known to regulate motivation, memory, and drug-reward circuits including the mPFC, dCPu, NAc, BLA, IPN, VTA, and locus coeruleus (LC) in naïve adolescent and adults male rats engaged in nicotine-seeking behavior containing the *CHRNA*6 3′-UTR SNP [29,30]. During adolescence, the α6 mRNA expression peaks [31]. In our study, animals undergoing nicotine + cue-primed reinstatement initiate nicotine during adolescence and face a reinstatement challenge in early adulthood. During this period, the brain undergoes maturational changes, including alterations in functional connectivity. These changes contribute to the development of executive function and cognitive control, which may be partly attributed to the maturation of the DA system. Assessing both naïve and nicotine-seeking rats will allow us to understand the baseline conditions and drug-induced effects aiding in identifying specific mechanisms driving drug-seeking behavior in the *CHRNA*6 3′-UTR SNP knock-in rats. We hypothesize that the *CHRNA*6 3′-UTR SNP will alter DA, NE, and DA metabolite levels, and their turnover rates in limbic brain regions in an age-, sex-, and genotype-dependent manner, with alterations in neurotransmitter levels predicting nicotine-seeking behavior [32,33,34,35,36].

## 2. Results

### 2.1. Comparative DA, NE, and Metabolite Profile Analysis of Adolescent and Adult, α6^GG^ and α6^CC^ in Distinct Limbic Brain Regions

Adolescent (PN 32) and adult (PN 60) male and female, α6^GG^ and α6^CC^, and tissue level NE, DA, and DA metabolites were assessed in key regions of the reward circuitry. We observed age, genotype, and sex differences specific to each region analyzed (Figure 1A–N). Appendix A shows the non-significant results in each brain region assessed in the *CHRNA*6 3′-UTR SNP knock-in rat line. In the mPFC, a significant three-way interaction for Age × Genotype × Sex [F(1,66) = 5.4409, *p* = 0.0227] was observed for NE tissue levels. The data were separated by sex, genotype, and age. Both adult female α6^CC^ and α6^GG^ exhibited greater NE compared to adolescent female α6^CC^ and α6^GG^, respectively. Meanwhile, adolescent α6^GG^ and adult α6^CC^ males showed greater NE when compared to adolescent α6^CC^, * *p* < 0.05 and ** *p* < 0.01, respectively. The DA tissue level in the mPFC revealed an interaction for Age × Genotype × Sex [F(1,69) = 9.0890, *p* = 0.0036]. The post hoc analysis showed that adolescent α6^GG^ (*p* < 0.05) and adult α6^CC^ (*p* < 0.001) males exhibited greater DA levels compared to adolescent α6^CC^ males.

In the dCPu, a region known to be influenced by smoking-cue reactivity [37,38,39], the results showed a three-way interaction for Age × Genotype × Sex for DA [F(1,69) = 10.7272, *p* = 0.0017]. Adult α6^GG^ females (*p* < 0.01) exhibited greater DA compared to adolescent α6^GG^ females, and adult α6^CC^ males had greater DA compared to adolescent α6^CC^ (*p* < 0.001) and adult α6^GG^ (*p* < 0.05) males. 

In the BLA, an interaction for Age × Genotype × Sex [F(1,69) = 4.9885, *p* = 0.0288] for tissue NE was found with adult α6^GG^ females showing greater NE compared to adolescent α6^GG^ females (*p* < 0.05). Adult α6^CC^ males had greater NE levels compared to adolescent α6^CC^ (*p* < 0.01) and adult α6^GG^ (*p* < 0.05). Furthermore, the DA tissue level was influenced by Age × Genotype × Sex [F(1,69) = 4.9885, *p* = 0.0288], and the post hoc analysis revealed adult α6^GG^ females exhibited greater DA compared to adolescent α6^GG^ females (*p* < 0.001). In the NAc core an Age × Genotype × Sex interaction was observed [F(1,69) = 14.8104, *p* = 0.0003] with adolescent α6^GG^ and adult α6^CC^ males exhibiting greater DA compared to adolescent α6^CC^ and adult α6^GG^ (*p* < 0.05).

In the LC, an interaction for Age × Genotype × Sex for tissue NE [(F(1,69) = 7.7562, *p* = 0.0069] was observed. The post hoc analysis revealed that adolescent α6^GG^ males (*p* < 0.05) exhibited greater NE compared to adolescent α6^CC^ and adult α6^GG^ males. 

DOPAC/DA turnover differences were observed in the mPFC and LC. In the mPFC, the main effects of Sex [F(1,67) = 6.9398, *p* = 0.0105], a Sex × Age interaction [F(1,67) = 5.9179, *p* = 0.0177], and an Age × Genotype × Sex interaction [F(1,67) = 5.5082, *p* = 0.0219] were observed. Adult females α6^GG^ exhibiting a greater DOPAC/DA turnover was compared with adolescent α6^GG^ females in the mPFC and LC (*p* < 0.001 and *p* < 0.001, respectively). An Age × Genotype [F(1,64) = 6.5805, *p* = 0.0127] and Age × Genotype × Sex interaction [F(1,64) = 4.3568, *p* = 0.0408] for the HVA/DA turnover were discovered such that adult α6^GG^ females exhibiting a greater HVA/DA turnover was compared to adolescent α6^GG^ and adult α6^CC^ females (*p* < 0.05). In the LC, as the DOPAC/DA main effect for Age [F(1,68) = 26.7020, *p* = 0.0001], Genotype × Sex [F(1,68) = 6.8299, *p* = 0.0110] and Age × Genotype × Sex [F(1,68) = 7.2025, *p* = 0.0091] interactions were observed. Adult α6^CC^ males exhibited a greater DOPAC/DA ratio in the LC compared to adult α6^GG^ (*p* < 0.01) and adolescent α6^CC^ (*p* < 0.05). 

### 2.2. Sex- and Genotype-Dependent Effects on Tissue Neurotransmitter Levels in Rats Tested for Nicotine plus Cue-Induced Reinstatement

Our prior published findings demonstrate an impact on males to nicotine combined with cue reinstatement therefore, our attention was exclusively directed to males. In our published results, we show that male α6^GG^ exhibit enhanced nicotine + cue-seeking behavior when compared to males α6^CC^ (*p* < 0.05) [28]. In our current studies, similar to our published results [28], we observed no genotype effects for food acquisition, nicotine self-administration, the breakpoint for the progressive ratio, and extinction (Appendix A). When we evaluated nicotine + cue reinstatement responding (Figure 2A), an overall ANOVA revealed a main effect for the Genotype [F(1,14) = 9.6181, *p* = 0.0078], Reinstatement stimulus [F(1,14) = 24.6883, *p* = 0.0002], and Reinstatement stimulus × Genotype [F(1,14) = 7.07485, *p* = 0.0187]. Similar to our published results [28], the post hoc analysis further identified nicotine + cue-induced reinstatement in α6^GG^ males when compared to α6^CC^ males [F(1,14) = 10.1195, *p* = 0.0067] (Figure 2A). When assessing DA, NE, and metabolites or the turnover rates in nicotine + cue-induced reinstatement *CHRNA*6 3′-UTR SNP knock-in male rats, we did not observe any genotype or interactive effects (*p* > 0.05). 

Subsequently, we evaluated any change in DA, NE, metabolites, or turnover rates between male naïve adolescents, adults, and reinstatement *CHRNA*6 3′-UTR SNP knock-in rats in key regions of the reward pathway. An overall ANOVA revealed the main effects for the Condition [F(2,42) = 4.8966, *p* = 0.0123] and Condition × Genotype [F(2,42) = 3.6459, *p* = 0.0347]. The post hoc analysis revealed α6^GG^ nicotine + cue reinstatement males had significantly less DA levels compared to naïve adolescent (*p* = 0.0009) and adult (*p* = 0.0329) males in the NAc shell, as shown in Figure 3. In order to identify possible alterations in the tissue DA metabolism in animals containing the *CHRNA*6 3′-UTR SNP, we assessed the DA metabolite turnover. An overall ANOVA revealed a main effect for the condition [F(2,42) = 13.8813, *p* = 0.0001]. The post hoc analysis revealed an increased in the HVA/DA turnover for nicotine-seeking α6^GG^ males in the NAc core and shell, BLA, and VTA when compared to both, adolescents and adult α6^GG^ males, but not in α6^CC^ males (Table 1). Furthermore, nicotine-seeking male α6^GG^ and α6^CC^ exhibited changes in HVA/DA in the mPFC, dCPu, IPN, and LC when compared to adolescent and adult naïve *CHRNA*6 3′-UTR SNP knock-in rats (Table 1). The DOPAC/DA turnover is increased in α6^GG^ and α6^CC^ males independent of the genotype after reinstatement testing in the NAc shell. In the IPN, DOPAC/DA is increased in nicotine-seeking α6^CC^ males, but not in nicotine-seeking α6^GG^ males when compared to adolescents and adults at the baseline (Appendix A). No other differences for DOPAC/DA were found.

We explored the relationship between the DA, NE, and DA metabolite content and behavioral responses to nicotine + cues, focusing on specific correlations within α6^GG^ male rats. There was a positive correlation for NE in the LC (Figure 2B) and DA in the BLA (Figure 2C), whereas a negative correlation for DA in the VTA (Figure 2D) was observed for α6^GG^ males, but not α6^CC^ males. Moreover, we detected a positive correlation for the HVA/DA turnover and behavioral response in the α6^GG^ mPFC (Figure 2E). Conversely, a negative correlation between the HVA/DA turnover and nicotine + cue response was found in the LC in α6^GG^ males (Figure 2F). 

## 3. Discussion

Previous studies have extensively explored the role of the *CHRNA*6 3′-UTR SNP in acute versus sub-chronic nicotine-induced behaviors and nicotine + cue-primed reinstatement [27,28]. These studies revealed that the effects are genotype- and sex-dependent with a greater impact in α6^GG^ males and α6^CC^ females [27,28]. The present studies illustrate that naïve adolescent, adult, and nicotine + cue reinstatement DA, NE, and DA metabolite levels and turnover rates in the key brain regions of the reward circuitry are influenced by age, genotype, and sex in the *CHRNA*6 3′-UTR SNP knock-in rats. Our study found elevated levels of DA in the mPFC and LC, as well as NE levels in the mPFC, NAc core, and LC in naïve adolescent α6^GG^ males, while adult α6^CC^ males showed higher DA in the mPFC, dCPu, and BLA, along with increased NE in the mPFC and NAc core. Conversely, naïve adult α6^GG^ females showed elevated tissue DA levels in the BLA and NE in the mPFC, dCPu, and BLA, and adolescent α6^CC^ females exhibited higher NE levels in the mPFC. Additionally, we successfully replicated previous findings on nicotine-seeking behavior in male rats with the *CHRNA*6 3′-UTR SNP. When evaluating the DA levels in adolescents, adults, and α6^CC^ and α6^GG^ males exposed to nicotine + cue, we observed a notable decrease in DA levels in the NAc shell, particularly in α6^GG^ males. Additionally, α6^GG^ males exhibited an increased HVA/DA turnover in key regions related to the reinforcing effects of nicotine. Pearson correlation analyses revealed a positive correlation between DA levels in the BLA and NE levels in the LC, predicting an enhanced nicotine + cue response in α6^GG^ males. Conversely, a negative correlation in the VTA of α6^GG^ males predicted a greater behavioral response to nicotine-seeking rats. Furthermore, higher HVA/DA in the mPFC and a reduced HVA/DA turnover in the LC were predictive of increased nicotine + cue behavior in α6^GG^ males. Collectively, these results suggest that the levels of DA, NE, and DA metabolites undergo developmental changes, mediated by α6 nAChRs. These processes are further impacted by the human genetic polymorphism *CHRNA*6 3′-UTR SNP. 

Our results provide initial evidence that the HVA/DA turnover tissue levels in the mPFC and LC may serve as predictors for reinstatement behavior involving the potential α6 nicotinic receptor subunit in *CHRNA*6 3′-UTR SNP male rats. Previous research in adult male Sprague Dawley rats receiving nicotine doses (3 or 12 mg/kg/day) via an osmotic minipump observed a decreased DOPAC+HVA/DA turnover, primarily decreased in DOPAC, with less of an effect on HVA in the striatum [40]. In contrast, our study revealed a greater HVA/DA turnover in α6^GG^ males, but not in α6^CC^ males in key regions of the reward pathway in nicotine-seeking *CHRNA*6 3′-UTR SNP knock-in rats. An elevated HVA/DA ratio in the α6^GG^ suggests that an increase in the extracellular DA metabolism can have significant clinical implications for understanding and potentially predicting the relapse behavior in individuals with this genotype. Biomarkers related to the α6 nicotinic receptor subunit activity and DA levels in the mPFC and LC could be used for personalized treatment plans. Tailoring interventions to an individual’s neurobiological profile might improve treatment outcomes. Further, our study examines differences between naïve adolescent and adult males in *CHRNA*6 3′-UTR SNP rats, which illustrate the baseline differences. Whether and how these noted baseline differences in DA, NE, and HVA/DA ratios relate to maturational effects that could potentially influence behavior require further investigation. 

Nicotine stimulates the firing rate of LC neurons, stimulating the release of NE and upregulating tyrosine hydroxylase mRNA, the rate limiting enzyme in the biosynthesis of catecholamines [41,42]. The release of NE from LC neurons into various brain regions, including the mPFC, where it enhances alertness and attention, and the amygdala, where it contributes to the stress response, can have implications in drug addiction and dependence [43]. Naïve adolescents α6^GG^ and adult α6^CC^ exhibited a greater NE tissue level compared to their counterparts in the mPFC, BLA, and LC, while these effects do not persist when α6^GG^ and α6^CC^ males become engaged in nicotine-seeking behavior. These results suggest that nicotine-seeking behavior may modulate or equalize NE levels in these brain regions, highlighting the influence of drug-seeking behavior on the neurochemical process in the *CHRNA*6 3′-UTR SNP rat line. Further, the LC-NE system can indirectly influence the activity of the DAergic system in the brain. NE and DA systems are interconnected, and they can modulate each other’s activity. LC-NE can influence the release of DA in certain brain regions, including the mPFC and the striatum [42]. During high arousal and/or stress, the interplay between NE and DA are critical factors, as they can work together to adapt the brain’s response to challenging situations. The stress response triggered by drug withdrawal or craving can activate the LC-NE system, leading to increased stress-related arousal and alertness. Simultaneously, the DA system can be activated as a response to the expectation of the rewarding effects of the drug, leading to motivation to seek the drug, even under stressful conditions. How NE and DA interplay are affected in *CHRNA*6 3′-UTR SNP knock-in rats needs further evaluation. 

Adolescence is a critical period for the development of the DAergic system and α6 has been shown to modulate DA release in the NAc [4]. The DA response to psychostimulants including nicotine, methamphetamine, and cocaine in detoxified dependent individuals and in animals is impaired [43]. Our investigations in *CHRNA*6 3′-UTR SNP knock-in rats showed decreased DA levels in the NAc shell of α6^GG^ males after reinstatement testing, a phenomenon not observed in α6^CC^ males. The dysregulation of DA function may motivate α6^GG^ males to seek drugs to restore normal DA levels, as observed by the elevated nicotine + cue reinstatement response in the α6^GG^ males as compared to the α6^CC^ males. In addition, the DA levels in the prelimbic (PL) region of the mPFC and the NAc shell, but not the core, have been shown to trigger reinstatement [44,45,46]. Extinction training, which involves learning new contextual relationships, may impact DA and DA receptors. Prior studies in male adult Wistar rats assessed cue-induced nicotine seeking during withdrawal, either with or without extinction training after nicotine self-administration, resulting in the decreased nicotine-seeking responding of the animals which underwent extinction [47]. Additionally, adult male Sprague Dawley rats’ administration of the D1 and D2 antagonists effectively attenuated the nicotine-seeking response elicited by the presentation of a previous nicotine-associated cue, suggesting the role of DAergic transmission [48]. Future studies are needed to understand the role of DA transmission in *CHRNA*6 3′UTR SNP knock-in rats after nicotine withdrawal and for an assessment of the functional DA release.

The clinical implications of the *CHRNA*6 3-UTR SNP were significantly associated with nicotine addiction and dependence phenotypes [18,20,21,24,49]. These include the number of unsuccessful quit attempts, primarily in males with the GG genotype as compared to their female counterparts [23]. Nicotine-exposed adolescents with the GG genotype have tried more cigarettes and drugs when compared to exposed C-carriers [18]. Numerous genetic studies have indicated a strong association between the *CHRNA*6 gene and an increased susceptibility to nicotine addiction and dependence. Nevertheless, the paucity of pharmacological tools to selectively target α6 nAChR makes it challenging to understand the mechanism of the variants of the α6 with nicotine addiction and dependence. Genetic manipulations, e.g., knock-out (KO) or knock-in (KI), have enabled investigators to bypass this issue [50,51]. α6-WT mice self-administered nicotine in a unit dose of 26.3 μg/kg/infusion (inf), whereas their α6-KO drug-naïve littermates did not [48]. The α6-KO animals did not self-administer nicotine even in an extensive range of lower (8.7–17.5 μg/kg/inf) and higher (35–52.6 μg/kg/inf) doses [48]. Notably, when the α6 subunit was selectively re-expressed in the VTA of α6^-/-^ mice using a lentiviral vector, the reinforcement property of nicotine was restored [48]. Further, α6 nAChR genetic KI strains have shown that a replacement of a Leu with Ser in the 9′ residue of the M2 domain of the α6 produces nicotine-hypersensitive mice (a6L9′s) with an enhanced DA release [52,53,54,55]. These α6L9′S strains show hyperactive locomotion; lacking habituation to their environment, such behaviors are consistent with enhanced DA neuron firing and release [52,53,54,55]. Furthermore, LC NE neurons likely contain α6β4* and α6β2* nAChR [16,56,57], suggesting *CHRNA6* may play a role in the release of NE. The mechanism by which the *CHRNA6* 3′-UTR SNP modulates LC NE neurons remains unknown. Further research is needed to fully elucidate the mechanisms underlying the association between the *CHRNA6* 3′-UTR SNP, changes in DA and NE, and nicotine addiction and dependence.

Given that α6 mRNA expression peaks during adolescence [31] and nicotine dose-dependently affects sex-specific behaviors [58], it becomes essential to extend the evaluation of *CHRNA*6 3′-UTR SNP rats to the adult stage. Future studies are needed to investigate α6 mRNA expression in *CHRNA*6 3′-UTR SNP rats after nicotine + cue reinstatement to determine potential changes in the gene expression associated with drug-seeking behavior. While previous research has shown that adolescent nicotine sub-chronic exposure does not interact with sex or genotype to influence α6 mRNA expression in specific brains regions such as the VTA, SNg, and IPN in the *CHRNA*6 3′-UTR SNP rats [27], it is imperative to explore its effects in the context of nicotine + cue reinstatement. As a limitation, the current study excluded females after nicotine + cue reinstatement due to a lack of behavioral effects. Thus, future studies will need to evaluate DA, NE, and their metabolites in females, α6^GG^, and α6^CC^ and compare these effects to our current results.

## 4. Materials and Methods

### 4.1. Animals

Male and female human *CHRNA*6 3′-UTR^C123G^ SNP rats were designed and bred in-house, as described in Cardenas et al. and Carreño and Lotfipour [27,28]. Both males and females will be included in the naïve adolescent and adult studies since alterations in nAChR expression and/or function have been shown to induce sex-dependent behavioral effects. For behavioral testing, only males were assessed as no genotype-dependent effects were observed in our previous study in females which underwent reinstatement [28]. Rats will be grouped by house in a controlled AAALAC-accredited 50% and temperature environment (21 °C)-controlled vivarium. Food and water will be available except during self-administration and reinstatement paradigm. Animals will be held for three days before beginning experimental paradigms. All experimental procedures have been approved by the Institutional Animal Care and Use Committee of the University of California, Irvine.

### 4.2. Tissue Catecholamine Levels and HPLC-ED Detection in Naïve CHRNA6 3′-UTR SNP Knock-In Rats

Brain tissue from naïve animals, aged postnatal day (PN) 32 and PN 60 were immediately removed via decapitation, rapidly frozen in −20 °C 2-methylbutane and stored at −80 °C until use. Brain tissue was sectioned at 300 µm on a cryostat set to −12 °C (Leica, Deer Park, IL, USA) [11]. Brain sections taken contained the mPFC, dCPu, NAc, BLA, VTA, IPN, and LC, which were identified with a rat brain atlas [59]. These brain regions of the corticostriatal-limbic system were assessed due to their involvement in reward and motivated behaviors [32,33,34,35,36,59,60,61]. Brain sections were briefly frozen in dry ice before tissue samples were dissected bilaterally with a 1 mm-diameter tissue punch (Integra, Mansfield, MA, USA). Tissue was expelled into 300 μL of ice-cold 0.1 M perchloric acid and homogenized. Samples were centrifuged at 10,000× *g* for 10 min, and the resulting pellets were resuspended in 100 mL of 0.1 M NaOH overnight before measuring the protein content using a BCA protein assay kit (Pierce, Rockford, IL, USA). Protein content was quantified using a Fluorometer (Invitrogen, Waltham, MA, USA). The supernatants were used for the measurement of NE, DA, and metabolites using high-performance liquid chromatography coupled with electrochemical detection (HPLC-ECD). Samples on different brain regions were run on similar date, but samples from age differences were run on separate dates. Tissue sections were examined anatomically after tissue puncture to verify correct localization of tissue samples. Tissue samples were automatically injected by an ESA 542 refrigerated autosampler onto a 150 × 3.2 mm ODS C18 column (ESA Inc., Chelmsford, MA, USA) connected to an ESA 580 HPLC pump. The column was kept at 37 °C and perfused by MD-TM mobile phase (ThermoFisher, Waltham, MA, USA) at a rate of 0.6 mL/min. NE, DA, and metabolite levels were determined by an electrochemical ESA 5600 detected with an ESA 5020 guard cell with the dominant potential of 160 mV. The sensitivity of the detector is 500 fg. Measurements were analyzed using CoulArray for Windows software 2.0 (ESA Inc., Chelmsford, MA, USA). Standard curves were generated with catecholamines standard (ThermoFisher, Waltham, MA, USA), DOPAC, and HVA (Sigma-Aldrich, St. Louis, MO, USA) standards, and levels in experimental samples were determined from the curve and expressed as ng/g, adjusted for protein concentration [32].

### 4.3. Apparatus for Behavioral Testing

Animals were tested in plexiglass operant chambers (Med Associates, St Albans, VT, USA), equipped with two levers The required number of responses at the reinforced (Reinf) lever turned on a cue light over the lever, turned off the house light, and activated an externally mounted syringe pump that infused drug. During the infusion (5.6 s yielding 100 μL of solution) and timeout period (20 s) the cue light remained illuminated, and the house light remained off. After the timeout period, the house light turned on and signaled the availability of a reinforcer. Responses on the non-reinforced (NonReinf) lever were recorded but had no consequences [28,62].

### 4.4. Food Self-Administration

Male adolescents (PN 24) were trained twice per day in a 30 min session to lever press for food pellets (45 mg rodent purified diet; Bio-Serv, Frenchtown, NJ, USA) in lever-pressing operant testing chambers (Med Associates, St. Albans, VT, USA) based on prior studies [63,64], as previously described [28,62]. One wall of the chamber contained two levers, a cue light over each, and a house light. The right lever was assigned as the active (Reinf) lever and each response at which was rewarded with delivery of food. The left lever was inactive (NonReinf) and had no consequences but was recorded as a measure of nonspecific activity. The animals started at an FR1TO1 (fixed-ratio 1, 1 s timeout) schedule of reinforcement, followed by FR1TO10, FR2TO20, and finally, FR5TO20, progressing upon earning 35 reinforcers.

### 4.5. Surgery

Following successful acquisition of food training, rats were anesthetized with Equithesin (0.0035 mL/g body weight) and implanted with indwelling jugular vein catheters [65]. After surgery, rats were given the analgesic carprofen (5 mg/kg, subcutaneous). During the 2–3-day recovery period, catheters were flushed daily with heparinized saline solution (1 mL of 1000 units/mL heparin into 30 mL of bacteriostatic saline) to maintain patency. Catheter patency was tested for rapid (5–10 s) anesthesia by infusing propofol (5 mg/kg, i.v.) before and after the completion of self-administration experiments. Only animals showing rapid anesthesia were included in analyses.

### 4.6. Nicotine Intravenous Self-Administration and Extinction

Animals (PN 34) intravenously self-administered (IVSA) and continued through progressive ratio and extinction as previously described [28,62]. Animals (PN 34) intravenously self-administered (IVSA) nicotine (0.015 mg/kg/infusion) at an FR5 schedule for 1 h daily session for a minimum of 5 days, or until they reached stable responding (Reinforced responses (Reinf) within 20% of the mean over the last 3 days; 2 × Reinf ≥ NonReinforced responses; Reinf ≤ 5). A dose of 0.015 mg/kg/infusion was chosen based on previous adult and adolescent studies [28,62,63,66]. Baseline responding was defined as the average reinforced responses over the last three days of self-administration. Rats were then allowed to respond at the dose of 0.015 mg/kg/infusion on Progressive Ratio (PR) schedule (~PN 39). The PR of reinforcement is a measure of motivation to obtain the drug [67]. The sequence was determined using the exponential formula (5 exp (0.2 × infusion number) − 5) such that the required responses per infusion were as followed: 1, 2, 4, 6, 9, 12, 15, 20, 25, 32, 40, 50, 151 62, 77, 95, 118, 145, 178, 219, 268, 328, 402, and 492 [67]. PR conditions were the same for FR sessions, with the exception that the sessions were 4 h duration. Breakpoint was achieved when >20 min of inactivity on the active lever elapsed. After reaching stable responding and two days of PR schedule, extinction–reinstatement testing began. During extinction (~PN 41), animals were placed in the same operant testing chambers; the animals were not connected to the infusion tubing, the house light remained on and responses on the levers were counted but had no consequences.

### 4.7. Cue and Nicotine-Induced Reinstatement and Tissue Catecholamines Levels

After meeting extinction criteria, reinstatement testing began (~PN 47), as previously described [28,62]. In brief, nicotine seeking was reinstated using nicotine-primed paired with cue. Presentation of cue consisted of cue light illumination and sound in the testing chamber. Nicotine-prime injections contained 0.15 mg/kg nicotine and were administered intraperitoneally immediately before the reinstatement test. Upon completion of the reinstatement test, animals were quickly decapitated. Brain tissue was immediately removed, rapidly frozen in −20 °C 2-methylbutane and stored at −80 °C until use. Brain tissues were collected and processed as mentioned above for determination of catecholamines.

### 4.8. Statistical Analysis

Data were analyzed using JMP (SAS Institute) software version 17. All data are expressed as mean ± SEM. Each neurotransmitter, metabolite, and metabolite ratio were individually analyzed in each distinct brain regions. Tissue level neurotransmitters in naïve animals were analyzed by a three-way ANOVA for Age (adolescent and adult) × Sex (male and female) × Genotype (α6^GG^ and α6^CC^). Significant main effects were analyzed with 1-way ANOVAs and Bonferroni-corrected paired or unpaired t-tests, as appropriate. Food acquisition was analyzed by a compound 3-way multivariate ANOVA for Lever Presses × Genotype × FR schedule with repeated measures on Lever Presses and FR schedule, with Bonferroni-corrected *t*-test post hoc comparisons. Nicotine self-administration data were analyzed by a compound 3-way multivariate ANOVA Reinf/Nonreinf Responses × Day × Genotype day (day 3–5) with repeated measures on Reinf/Nonreinf Responses and Day [28]. Reinstatement data were analyzed as normalized reinforced responding [28]. Mean responses for nicotine + cue reinstatement were analyzed by a 2-way multivariate ANOVA for Genotype × Reinstatement Condition, with repeated measures on Reinstatement Condition [28]. Neurotransmitter data in animals assessed for reinstatement, as compared with adolescent and adult naïve rats, were evaluated by a 2-way ANOVA for Genotype × Condition (Adolescent, Adult, nic + cue). Significant main effects were further analyzed with 1-way ANOVAs and Bonferroni-corrected paired or unpaired *t*-tests, as appropriate. Pearson’s correlation coefficient was assessed to compare the DA, NE, DA metabolite levels, or turnover vs. nicotine + cue seeking behavior response, reporting the RSquare and *p*-values with a false discovery rate (FDR) of *p* < 0.05.

## 5. Conclusions

Our study sheds light on the intricate interplay between age, genotype, and sex in the *CHRNA*6 3′-UTR SNP knock-rats and its impact on DA, NE, and DA metabolite levels in key brain regions of the reward circuitry. We demonstrated the correlations of DA-, NE-, and DA-turnover-specific brain regions which can predict nicotine-seeking behavior during nicotine + cue reinstatement in α6^GG^ males. The levels of DA, NE, and DA turnover in these specific regions seem to play a role in modulating how *CHRNA*6 3′-UTR SNP rats respond to nicotine + cue drug-seeking behavior. Further research is needed to explore gene expression changes and immediate early gene responses during nicotine + cue reinstatement in this genetic rat model. Our findings contribute to the growing understanding of the neural underpinnings of nicotine addiction and may pave the way for future interventions targeting the α 6 nAChR subunit to combat nicotine addiction more effectively.

## Figures and Tables

**Figure 1 ijms-25-03676-f001:**
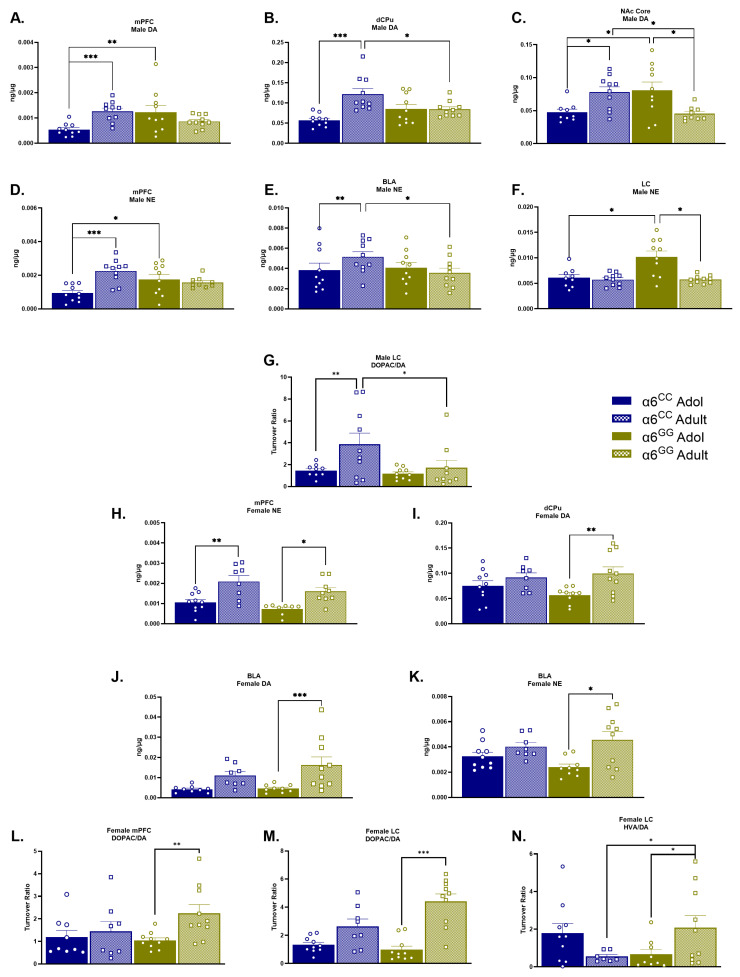
Sex and Genotype Differences in Naïve Adolescent (PN 32) and Adult (PN 60) DA, NE, and Turnover Profile in the Humanized *CHRNA*6 3′-UTR SNP rats. A complex interplay between sex, genotype, and sex in DA and NE regulation in males (**A**–**G**) and females (**H**–**N**). Adult α6^CC^ and adolescent (Adol) α6^GG^ show higher DA levels when compared to adolescent α6^CC^ males in the PFC (**A**) and NAc Core (**C**); while adult males α6^CC^ display increased DA levels when compared to adolescent α6^CC^ and adult α6^GG^ males in the dCPu (**B**). Within the NAc Core, adolescent α6^GG^ also exhibit greater DA when compared to adult α6^GG^. NE is also elevated in adult males α6^CC^ and adolescent α6^GG^ in the mPFC compared to their respective counterparts (**D**). Adult α6^CC^ exhibit greater NE when compared to adolescent α6^CC^ and adult α6^GG^ in the BLA (**E**). Adolescent α6^GG^ males display greater NE in the LC when compared to adolescent α6^CC^ and adult α6^GG^ males (**F**). Adult males α6^CC^ display greater DOPAC/DA turnover ratios when compared to adolescent α6^CC^ and adult α6^GG^ (**G**). On the contrary, adult α6^GG^ females show elevated NE and DA in the mPFC (**H**), dCPu (**I**), and BLA (**J**,**K**) when compared to adolescent α6^GG^ females. Adult α6^CC^ females show greater NE when compared to adolescent α6^CC^ females. Potential differences in DA metabolism are indicated by greater DOPAC/DA turnover ratio and were observed for adult females α6^GG^ when compared to adolescent α6^GG^ females in the mPFC and LC (**L**–**M**). Furthermore, in the LC, HVA/DA turnover was greater for adult α6^GG^ females when compared to adult α6^CC^ and adolescent α6^GG^ (**N**). mPFC = medial Prefrontal cortex, dCPu = dorsal Caudate putamen, NAc = Nucleus accumbens (Shell and Core), VTA Ventral tegmental area, IPN = Interpeduncular nucleus, LC = Locus coeruleus, circles = adolescents (Adol), squares = adults * *p* < 0.05, ** *p* < 0.01, *** *p* < 0.001 α6^GG^ vs. α6^CC^. All data presented as mean ± SEM. N = 8–10/group.

**Figure 2 ijms-25-03676-f002:**
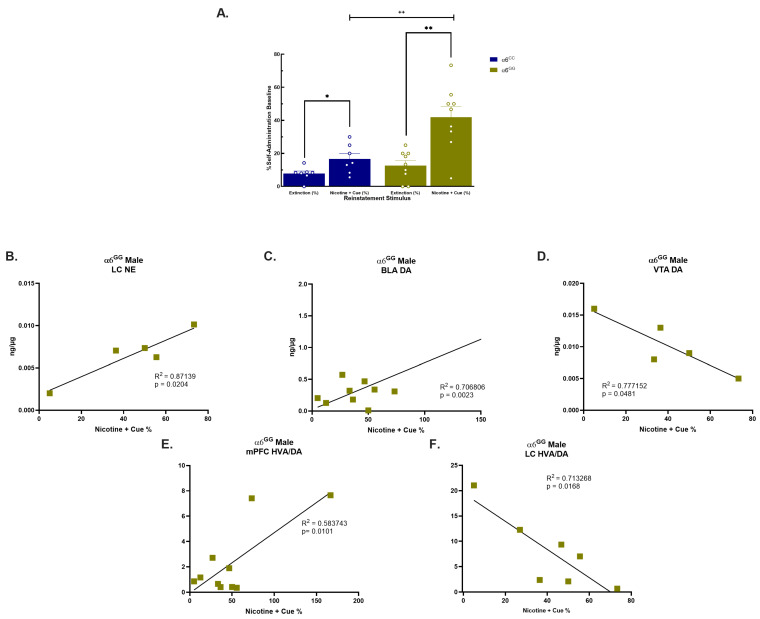
Correlations Between NE, DA, and HVA/DA Turnover Ratio and Reinstatement Behavior in the *CHRNA*6 3′-UTR males. (**A**) α6 3′-UTR SNP genotype dependently influences nicotine + cue-primed reinstatement, with α6^GG^ males more impacted than α6^CC^ males (~PN47). Data represent the mean (±SEM) of nicotine + cue-seeking responding in male, α6^GG^ vs. α6^CC^. * *p* < 0.05, ** *p* < 0.01 vs. Extinction; ++ *p* < 0.01 α6^GG^ vs. α6^CC^ Pearson correlation of NE, DA, and HVA/DA turnover males α6^GG^. Positive correlation between Nicotine + cue-primed reinstatement and increased NE levels in the LC (**B**) and DA levels in the BLA (**C**) in α6^GG^ males. On the contrary, a negative correlation between nicotine + cue-primed reinstatement and DA levels in the VTA (**D**). A lower HVA/DA turnover ratio in the mPFC and LC in α6^GG^ males suggests a greater nicotine + cue behavioral response (**E**,**F**). N = 5–9/group. * *p* < 0.05, ** *p* < 0.01 vs. Extinction; ++ *p* < 0.01 α6^GG^ vs. α6^CC^.

**Figure 3 ijms-25-03676-f003:**
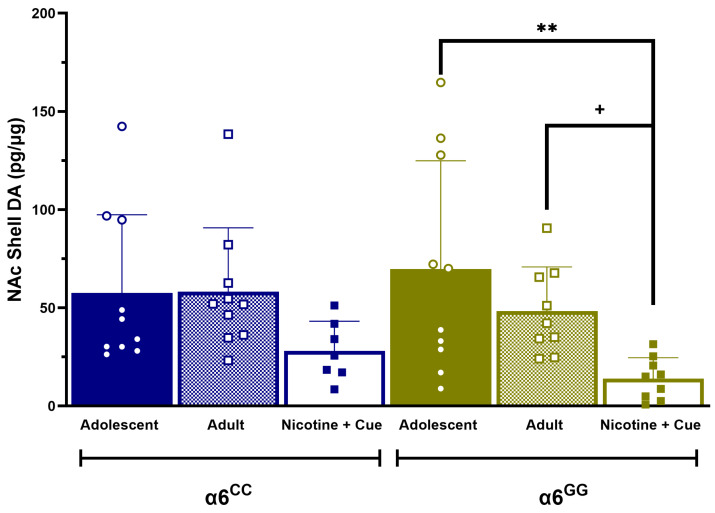
Age and genotype differences in the NAc Shell in the *CHRNA*6 3′-UTR SNP knock-in rats. DA (pg/μg) levels in the NAc shell, a brain region associated with reward, pleasure, and addition, are presented as the mean (±SEM) of naïve adolescents and adults, as well as nicotine + cue-seeking *CHRNA*6 3′-UTR SNP knock-in rats. Nicotine-seeking GG male rats exhibit substantially decreased DA levels when compared to naïve adolescent and adult *CHRNA*6 3′-UTR SNP knock-in rats. Open circles = adolescents; open squares = adults, and closed squares = reinstatement animals. N = 7–10. ** *p* < 0.01 vs. adolescents; + *p* < 0.05 vs. adults.

**Table 1 ijms-25-03676-t001:** HVA/DA Turnover in Key Regions of the Reward System in *CHRNA*6 3′-UTR SNP Knock-in Male Rats.

	HVA/DA					
	α6^CC^			α6^GG^		
	Adolescents	Adults	Reinstatement	Adolescents	Adults	Reinstatement
mPFC	0.63 ± 0.13	0.45 ± 0.12	3.81 ± 1.57 **^++^	0.65 ± 0.13	0.6 ± 0.12	2.86 ± 1.06 *^+^
dCPu	0.05 ± 0.01	0.07 ± 0.01	0.03 ± 0.01 ^++^	0.04 ± 0.01	0.07 ± 0.01	0.04 ± 0.00 ^++^
NAc Core	0.04 ± 0.01	0.09 ± 0.01	0.07 ± 0.01	0.03 ± 0.01	0.11 ± 0.01	0.06 ± 0.01 ^++^
NAc Shell	0.03 ± 0.01	0.04 ± 0.01	0.15 ± 0.11	0.05 ± 0.01	0.04 ± 0.01	0.29 ± 0.07 ***^+++^
BLA	0.14 ± 0.03	0.06 ± 0.03	0.31 ± 0.15	0.14 ± 0.03	0.1 ± 0.03	0.35 ± 0.10 ^++^
VTA	0.08 ± 0.02	0.1 ± 0.02	0.3 ± 0.33	0.13 ± 0.02	0.11 ± 0.02	0.62 ± 0.20 ***^+++^
IPN	1.22 ± 0.32	0.25 ± 0.33	5.27 ± 1.82 **^+++^	1.06 ± 0.32	0.38 ± 0.32	2.24 ± 0.86 ^+^
LC	1.08 ± 0.40	0.56 ± 0.40	5.27 ± 3.17 ^+^	1.16 ± 0.40	0.91 ± 0.40	6.86 ± 1.94 ***^+++^

Abbrev.: mPFC = medial prefrontal cortex; CPu = dorsal caudate putamen; NAc = nucleus accumbens; BLA = basolateral amygdala; VTA = ventral tegmental area; IPN = Interpeduncular nucleus; LC = locus coeruleus. * *p* < 0.05, ** *p* < 0.01; *** *p* < 0.001 vs adolescents; ^+^
*p* < 0.05, ^++^
*p* < 0.01, ^+++^
*p* < 0.001 vs. adults. N = 7–11.

## Data Availability

Data will be available upon reasonable request from the authors.

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
