# Peer review of "Dopamine and Norepinephrine Tissue Levels in the Developing Limbic Brain Are Impacted by the Human CHRNA6 3′-UTR Single-Nucleotide Polymorphism (rs2304297) in Ratsâ€"

_ijms, 2024, doi:10.3390/ijms25073676_

Round 1
Reviewer 1 Report
Comments and Suggestions for Authors
The use of chloral hydrate as an anesthetic in rodents has been widely questioned. You use a product that combines CH with another anesthetic (sodium pentobarbital). In your opinion, would it be possible to combine CH with an analgesic? Which would be the most appropriate?
Author Response
Reviewer 1:
- The use of chloral hydrate (CH) as an anesthetic in rodents has been widely questioned. You use a product that combines CH with another anesthetic (sodium pentobarbital). In your opinion, would it be possible to combine CH with an analgesic? Which would be the most appropriate?
There are risks associated with the use of chloral hydrate in rodents. Alternatives to sodium pentobarbital include a combination of ketamine and chloral hydrate, which is commonly used to anesthetize rats, mice, and other rodents. In our studies, we use a combination of chloral hydrate and sodium pentobarbital to replicate previous findings from our lab.
Reviewer 2 Report
Comments and Suggestions for Authors
IJMS-2908897-peer-review-v1.
England, 12th of March 2024.
The article by Carreño and colleagues examines the biochemical and behavioural consequences of a specific GG mutations in CHRNA6, coding for an alpha-6 nicotinic receptor. Authors found many significant tissue alterations of dopaminergic- and noradrenergic-related levels in several limbic brain regions. Furthermore, correlations were observed between biochemical measurements and behavioural performance during extinction.
Although well written, the manuscript, in its current form, will benefit from major revisions before it can be recommended for publication. Please see below for both major and minor comments.
Major comments:
-The similarity report is very high. Upon further examination, this article contains section included in the PhD thesis by Dr Carreño (UC Irvine, Pharmacological Sciences). Please include a statement to reflect this, similarly to what has been performed here: https://www.mdpi.com/1422-0067/23/15/8588.
-Please explain in more details the rationale surrounding tissue levels of DA, NE (and their metabolites) in relation to the CHRNA6 mutation. How could a mutation in a nicotinic receptor influence neurotransmitter levels? This is unclear.
-Pages are not numbered adequately, rending the reviewing process rather difficult.
-Section 5.2, lines 337-338. Please give additional details on how the brains were removed. What was the euthanasia technique?
-All figures: please display individual data points in addition to the current histograms.
-Figure 2, panel A. Here, authors represent 3 post-hoc tests. However, only two make clear sense. What is the horizontal bar representing? Are there missing elements here? I do not understand what the third comparison related to. Please explain, as the legend is also very confusing, mentioning difference between CC and GG, whilst the horizontal bar clearly encompasses GG only.
-Figure 2, panels B and C. Here, authors have n=5 (panel B) and n=9 (panel C), but surely the same animals were used to quantify tissue levels of NE (LC) and DA (BLA). Authors need to explain such inconsistencies in the number of values reported in data (and thus analyzed). Similar comment for panels D to F.
-Supplementary Figure 2, panels A and B. The dashed lines have no associated legends, or the legend is incorrect. At present, readers would not be able to distinguish the 4 groups from one another. Please explicit in the displayed legend which groups correspond to the 4 lines represented.
-A summary diagram on differential tissue expression of DA, NE and metabolites would be helpful.
-The discussion does not address how mutations in CHRNA6 can lead to altered levels of DA, NE and metabolites. Please discuss the possible molecular mechanisms involved. Please see above another comment relating to this (second in the list).
Minor comments:
-Page 1, lines 41-42. Please edit the sentence, as it is grammatically incorrect (facilitating versus facilitates).
-Page 2, line 49. Please correct the typo (unstranslated versus untranslated).
-Section 5.1, line 325. Here, two studies are cited, but with only one reference (number 17). Please edit.
-Section 5.2, lines 356-357. Please give further details on the composition of the MD-TM mobile phase.
-Section 5.3, line 367 is missing a full stop.
-Throughout: please use consistent in-text citation format. Indeed, some references are (x), others [x].
Author Response
The article by Carreño and colleagues examines the biochemical and behavioural consequences of a specific GG mutations in CHRNA6, coding for an alpha-6 nicotinic receptor. Authors found many significant tissue alterations of dopaminergic- and noradrenergic-related levels in several limbic brain regions. Furthermore, correlations were observed between biochemical measurements and behavioural performance during extinction.
Although well written, the manuscript, in its current form, will benefit from major revisions before it can be recommended for publication. Please see below for both major and minor comments.
Major comments:
- The similarity report is very high. Upon further examination, this article contains section included in the PhD thesis by Dr Carreño (UC Irvine, Pharmacological Sciences). Please include a statement to reflect this, similarly to what has been performed here: https://www.mdpi.com/1422-0067/23/15/8588.
The manuscript is part of Dr. Diana Carreño’s dissertation, and we have included a statement indicating this.
- Please explain in more detail the rationale surrounding tissue levels of DA, NE (and their metabolites) in relation to the CHRNA6 mutation. How could a mutation in a nicotinic receptor influence neurotransmitter levels? This is unclear.
We have added a statement in the introduction to explain the neuronal a6 subunits are expressed on DA-releasing neurons in the brain including LC neurons, a source of noradrenergic innervations in the brain. Any possible mutations could alter neurotransmitter response.
- Pages are not numbered adequately, rending the reviewing process rather difficult.
The manuscript was formatted to accommodate the journal requirements.
- Section 5.2, lines 337-338. Please give additional details on how the brains were removed. What was the euthanasia technique?
The brains were removed via decapitation, and we have added a statement to indicate this.
- All figures: please display individual data points in addition to the current histograms.
We have formatted all figures to show the the individual data points.
- Figure 2, panel A. Here, authors represent 3 post-hoc tests. However, only two make clear sense. What is the horizontal bar representing? Are there missing elements here? I do not understand what the third comparison is related to. Please explain, as the legend is also very confusing, mentioning the difference between CC and GG, whilst the horizontal bar clearly encompasses GG only.
For Figure 2A, we represent a 2-way ANOVA genotype (a6CC vs a6GG) x reinstatement stimulus (extinction vs nicotine + cue) given that we are only evaluating males. The legend contains a6CC(blue) vs a6GG (green). The x-axis represents the reinstatement condition, (extinction and nicotine + cue).
- Figure 2, panels B and C. Here, authors have n=5 (panel B) and n=9 (panel C), but surely the same animals were used to quantify tissue levels of NE (LC) and DA (BLA). Authors need to explain such inconsistencies in the number of values reported in data (and thus analyzed). Similar comment for panels D to F.
We collected data for all the animals assessed for reinstatement, however due to the posterior nature of the LC, we were only able to collect LC from 5 a6GG animals.
- Supplementary Figure 2, panels A and B. The dashed lines have no associated legends, or the legend is incorrect. At present, readers would not be able to distinguish the 4 groups from one another. Please be explicit in the displayed legend which groups correspond to the 4 lines represented.
We have corrected the legends on Supplemental Figure 1 A and B.
- A summary diagram on differential tissue expression of DA, NE and metabolites would be helpful.
We have included a supplemental table 1 depicting an overall summary of naïve adolescent and adults a6CC and a6GG.
- The discussion does not address how mutations in CHRNA6 can lead to altered levels of DA, NE and metabolites. Please discuss the possible molecular mechanisms involved. Please see above another comment relating to this (second in the list).
In the discussion, we have included a statement addressing the notable link between the CHRNA6 gene and nicotine addiction phenotypes, as observed in both clinical and preclinical studies. Our findings, combined with these studies, suggest that DA, NE, and metabolites may respond to variations in the CHRNA6 gene. This implies that mutations in this gene could potentially alter neurotransmitter response.
Minor comments:
- Page 1, lines 41-42. Please edit the sentence, as it is grammatically incorrect (facilitating versus facilitates).
We have inserted the grammatically correct word.
- Page 2, line 49. Please correct the typo (unstranslated versus untranslated).
We have corrected the typo.
- Section 5.1, line 325. Here, two studies are cited, but with only one reference (number 17). Please edit.
We have inserted the additional reference.
- Section 5.2, lines 356-357. Please give further details on the composition of the MD-TM mobile phase.
We have added further details of the distributor.
- Section 5.3, line 367 is missing a full stop.
We have added a full stop.
- Throughout: please use consistent in-text citation format. Indeed, some references are (x), others [x].
We have corrected in-text citation format.
Round 2
Reviewer 2 Report
Comments and Suggestions for Authors
Dear authors,
Thank you for addressing all of my comments. The manuscript can now be recommended for publication. Please note that typesetters will need to display page numbers consistently, as written in my earliest review (page numbering is inconsistent, as it currently displays from 1/17 to 9/17, then 2/17 to 9/17).
Best Regards
Comments on the Quality of English LanguageVery minor